# High-quality ultrastructural preservation using cryofixation for 3D electron microscopy of genetically labeled tissues

**Tin Ki Tsang[1†], Eric A Bushong[2†], Daniela Boassa[2], Junru Hu[2], Benedetto Romoli[3], Sebastien Phan[2], Davide Dulcis[3], Chih-Ying Su[1]\*, Mark H Ellisman[2,4]\***

[1]Neurobiology Section, Division of Biological Sciences, University of California, San Diego, La Jolla, United States; [2]National Center for Microscopy and Imaging Research, Center for Research in Biological Systems, University of California, San Diego, La Jolla, United States; [3]Department of Psychiatry, School of Medicine, University of California, San Diego, La Jolla, United States; [4]Department of Neurosciences, School of Medicine, University of California, San Diego, La Jolla, United States

**Abstract** Electron microscopy (EM) offers unparalleled power to study cell substructures at the nanoscale. Cryofixation by high-pressure freezing offers optimal morphological preservation, as it captures cellular structures instantaneously in their near-native state. However, the applicability of cryofixation is limited by its incompatibility with diaminobenzidine labeling using genetic EM tags and the high-contrast *en bloc* staining required for serial block-face scanning electron microscopy (SBEM). In addition, it is challenging to perform correlated light and electron microscopy (CLEM) with cryofixed samples. Consequently, these powerful methods cannot be applied to address questions requiring optimal morphological preservation. Here, we developed an approach that overcomes these limitations; it enables genetically labeled, cryofixed samples to be characterized with SBEM and 3D CLEM. Our approach is broadly applicable, as demonstrated in cultured cells, *Drosophila* olfactory organ and mouse brain. This optimization exploits the potential of cryofixation, allowing for quality ultrastructural preservation for diverse EM applications.
DOI: https://doi.org/10.7554/eLife.35524.001

**\*For correspondence:**
c8su@ucsd.edu (C-YS);
mark@ncmir.ucsd.edu (MHE)

[†]These authors contributed equally to this work

**Competing interests:** The authors declare that no competing interests exist.

## Introduction

The answers to many questions in biology lie in the ability to examine the relevant biological structures accurately at high resolution. Electron microscopy (EM) offers the unparalleled power to study cellular morphology and structure at nanoscale resolution (*Leapman, 2004*). Cryofixation by high-pressure freezing (hereafter referred to as cryofixation) is the optimal fixation method for samples of thicknesses up to approximately 500 μm (*Dahl and Staehelin, 1989*; *McDonald, 1999*; *Moor, 1987*; *Shimoni et al., 1998*). By rapidly freezing the samples in liquid nitrogen (−196 °C) under high pressure (~2100 bar), cryofixation immobilizes cellular structures within milliseconds and preserves them in their near-native state. In contrast, cross-linking-based chemical fixation takes place at higher temperatures (≥4 °C) and depends on the infiltration of aldehyde fixatives, a process which takes seconds to minutes to complete. During chemical fixation, cellular structures may deteriorate or undergo rearrangement (*Korogod et al., 2015*; *Steinbrecht and Müller, 1987*; *Szczesny et al., 1996*) and enzymatic reactions can proceed (*Kellenberger et al., 1992*; *Sabatini et al., 1963*), potentially resulting in significant morphological artefacts.

Cryofixation is especially critical, and often necessary, for properly fixing tissues with cell walls or cuticles that are impermeable to chemical fixatives, such as samples from yeast, plant, *C. elegans*,

and *Drosophila* (*Ding et al., 1993*; *Doroquez et al., 2014*; *Kaeser et al., 1989*; *Kiss et al., 1990*; *McDonald, 2007*; *Müller-Reichert et al., 2003*; *Shanbhag et al., 1999*; *Shanbhag et al., 2000*; *Winey et al., 1995*). As cryofixation instantaneously halts all cellular processes, it also provides the temporal control needed to capture fleeting biological events in a dynamic process (*Hess et al., 2000*; *Watanabe et al., 2013a*, *2013b*, *2014b*).

Despite the clear benefits of cryofixation, it is incompatible with diaminobenzidine (DAB) labeling reactions by genetic EM tags. For example, APEX2 (enhanced <u>a</u>scorbate <u>per</u>oxidase) is an engineered peroxidase that catalyzes DAB reaction to render target structures electron dense (*Lam et al., 2015*; *Martell et al., 2012*). Despite the successful applications of APEX2 to three-dimensional (3D) EM (*Joesch et al., 2016*), there has been no demonstration that APEX2 or other genetic EM tags can be activated following cryofixation. Conventionally, cryofixation is followed by freeze-substitution (*Steinbrecht and Müller, 1987*), during which water in the sample is replaced by organic solvents. However, the resulting dehydrated environment is incompatible with the aqueous enzymatic reactions required for DAB labeling by genetic EM tags.

EM structures can also be genetically labeled with fluorescent markers through correlated light and electron microscopy (CLEM). Yet, performing CLEM with cryofixed samples also presents challenges. Fluorescence microscopy commonly takes place either before cryofixation (*Brown et al., 2009*; *Kolotuev et al., 2009*; *McDonald, 2009*) or at a later stage after the sample is embedded (*Kukulski et al., 2011*; *Nixon et al., 2009*; *Schwarz and Humbel, 2009*). However, if the specimen is dissected from live animals, the time taken to acquire fluorescence images delays cryofixation and could cause ultrastructural deterioration. In order for fluorescence microscopy to take place after embedding, special acrylic resins need to be used (*Kukulski et al., 2011*; *Nixon et al., 2009*; *Schwarz and Humbel, 2009*) and only a low concentration of osmium tetroxide stain can be tolerated (*de Boer et al., 2015*; *Watanabe et al., 2011*). Although one can in principle perform fluorescence microscopy in cryofixed samples after rehydration, fluorescence images have only been acquired in sucrose-infiltrated cryosections (300–500 nm) (*Ripper et al., 2008*; *Stierhof and El Kasmi, 2010*). Moreover, no protocol has been developed to prepare large cryofixed tissues expressing genetic CLEM markers for high-contrast EM imaging. These constraints limit the applicability of CLEM for cryofixed samples.

Another disadvantage of cryofixation is that *en bloc* staining during freeze-substitution is often inadequate. As a result, post-staining of ultramicrotomy sections is frequently needed for cryofixed samples (*Shanbhag et al., 1999*; *Shanbhag et al., 2000*; *Takemura et al., 2013*). However, post-staining could be labor-intensive and time-consuming, especially for volume EM (*Ryan et al., 2016*; *Zheng et al., 2017*). Critically, on-section staining is impossible for samples imaged with block-face volume EM techniques (*Briggman and Bock, 2012*), such as serial block-face scanning electron microscopy (SBEM) (*Denk and Horstmann, 2004*). A large amount of heavy metal staining is necessary for SBEM to generate sufficient back-scatter electron signal and to prevent specimen charging (*Deerinck et al., 2010*; *Kelley et al., 1973*; *Tapia et al., 2012*). Therefore, it remains impossible to image cryofixed samples with SBEM or other techniques that require high-contrast staining.

To overcome these limitations of cryofixation, here we present a robust approach, named the CryoChem Method (CCM), which combines key advantages of cryofixation and chemical fixation. This technique enables labeling of target structures by genetically encoded EM tags or fluorescent markers in cryofixed samples, and permits high-contrast *en bloc* heavy metal staining sufficient for SBEM. Specifically, we rehydrate cryofixed samples after freeze-substitution to make the specimen suitable for subsequent aqueous reactions and fluorescence imaging. We successfully apply CCM to multiple biologically significant systems with distinct ultrastructural morphology, including cultured mammalian cells, *Drosophila* olfactory organ (antenna) and mouse brain. By overcoming critical technical barriers, our method exploits the potential of cryofixation, making it compatible with genetically encoded EM tags and any EM techniques that require substantial heavy metal staining. Furthermore, the versatility of CCM allows us to achieve 3D CLEM in a well-preserved mouse brain by permitting SBEM after fluorescent imaging of a frozen-rehydrated specimen.

## Results

Given that a key limitation of cryofixation arises from the dehydrated state of the samples after freeze-substitution (*Table 1*), it is imperative that our approach delivers a cryofixed specimen that is

**Table 1.** Comparison of the advantages and limitations of different sample preparation methods for electron microscopy. The CryoChem Method (CCM) combines the advantages of chemical fixation and cryofixation. With CCM, samples are fixed with high-pressure freezing and freeze-substitution to achieve quality ultrastructural preservation. This approach allows preservation of tissues with cuticle or cell wall and captures biological events with high temporal resolution. A rehydration step is introduced to enable fluorescence imaging, DAB labeling by genetically encoded EM tags and high-contrast *en bloc* heavy metal staining of cryofixed samples. The high-contrast *en bloc* heavy metal staining permitted by CCM reduces the need for post-staining on sections, and makes CCM compatible with serial block-face scanning electron microscopy (SBEM). Common limitations of chemical fixation and cryofixation are denoted in red.

| | Chemical fixation | Cryofixation | CryoChem |
|---|---|---|---|
| Fixation | Aldehyde fixatives (4 °C) | (1) High-pressure freezing (−196 °C,~2100 bar) (2) Freeze-substitution in organic solvents | (1) High-pressure freezing (−196 °C,~2100 bar) (2) Freeze-substitution in organic solvents |
| Ultrastructural preservation | Fair | Excellent | Excellent |
| Tissues with cuticle or cell wall | Incompatible | Compatible | Compatible |
| Temporal resolution of events captured | Low | High | High |
| Hydration state of the sample | Hydrated | Dehydrated after freeze-substitution | Hydrated after rehydration |
| Fluorescence imaging after fixation | Compatible | Generally incompatible | Compatible |
| DAB labeling by genetic EM tags | Compatible | Incompatible due to dehydration | Compatible due to rehydration |
| High-contrast *en bloc* heavy metal staining | Compatible | Limited | Compatible due to rehydration |
| Post-staining on sections | Optional | Often required | Optional |
| SBEM compatibility | Compatible | Incompatible | Compatible |

DOI: https://doi.org/10.7554/eLife.35524.002

fully hydrated and can then be processed at higher temperatures (4 °C or room temperature) for enzymatic reactions and/or high-contrast *en bloc* heavy metal staining. It has been demonstrated that cryofixed samples can be rehydrated for immunogold labeling or fluorescence imaging following cryosectioning (*Dhonukshe et al., 2007*; *Ripper et al., 2008*; *Stierhof and El Kasmi, 2010*; *van Donselaar et al., 2007*), but these approaches only yield modest EM contrast. In addition, the methods are incompatible with volume EM techniques and have yet to be successfully combined with genetic labeling using APEX2.

## The CryoChem method

To achieve the ultrastructural preservation of cryofixation and the versatility of chemical fixation, we developed a hybrid protocol which we refer to hereafter as the CryoChem Method (CCM) (*Table 1*). Importantly, we devised a freeze-substitution cocktail (see below) that allows preservation of APEX2 enzymatic activity and signals from fluorescent proteins. CCM begins with high-pressure freezing of a sample, followed by freeze-substitution in an acetone solution with glutaraldehyde (0.2%), uranyl acetate (0.1%), methanol (2%) and water (1%), to further stabilize the cryo-preserved structures at low temperatures. After freeze-substitution, the sample is rehydrated gradually on ice with a series of acetone solutions containing an increasing concentration of water or 0.1 M HEPES. Once completely rehydrated, the cryofixed sample is amenable for imaging with fluorescence microscopy, DAB labeling reactions using genetically encoded tags, and the high-contrast *en bloc* staining (e.g. osmium-thiocarbohydrazide-osmium and uranyl acetate) normally reserved only for chemically fixed samples. Afterwards, the samples is dehydrated through a series of ethanol solutions and acetone, then infiltrated with epoxy resin and cured using standard EM procedures. To minimize volume artefact, epoxy resin is chosen because it causes minimal tissue shrinkage during embedding (<2%) compared to other embedding media (*Kushida, 1962*). The resin-embedded sample can be sectioned or imaged directly with any desired EM technique (*Figure 1*, see Materials and methods for details).

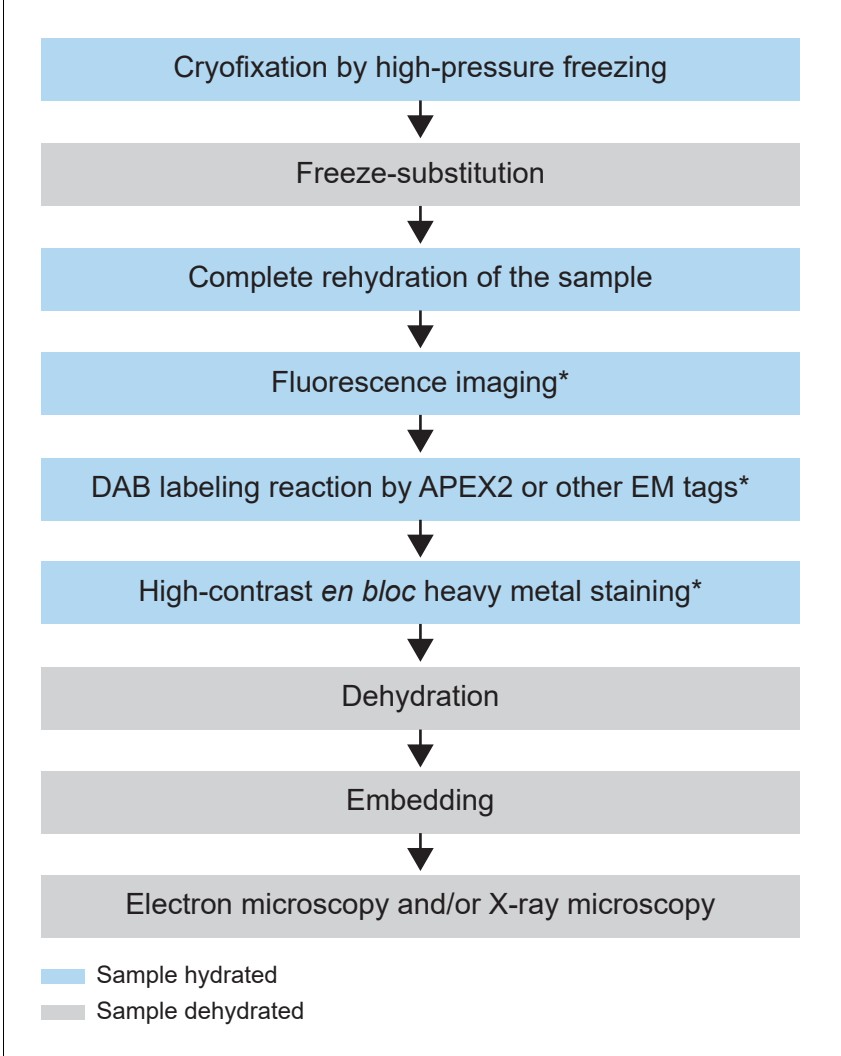

**Figure 1.** Flowchart of the CryoChem method. After cryofixation by high-pressure freezing and freeze-substitution, cryofixed samples are rehydrated gradually. Rehydrated samples can then be imaged for fluorescence, subjected to DAB labeling reaction or *en bloc* stained with a substantial amount of heavy metals. The protocol is modular; the first three processes are the core steps of CCM and the starred steps are optional depending on the experimental design. The samples are then dehydrated for resin infiltration and embedding, followed by imaging with any EM technique of choice. Blue and grey denote hydrated and dehydrated states of the sample, respectively.

DOI: https://doi.org/10.7554/eLife.35524.003

## CryoChem method offers high-quality ultrastructural preservation and sufficient *en bloc* staining for SBEM

To determine whether CCM provides high-quality ultrastructural preservation, we first tested the method in a mammalian cell line. Using transmission electron microscopy (TEM), well-preserved mitochondria and nuclear membrane were observed in the CCM-processed cells (*Figure 2—figure supplement 1*). Given that cryofixation is often necessary for properly fixing tissues surrounded by a barrier to chemical fixatives (*Steinbrecht, 1980*; *Steinbrecht and Müller, 1987*), we next tested CCM in a *Drosophila* olfactory organ, the antenna, which is encased in a waxy cuticle (*Figure 2A and B*). A hallmark of optimally preserved antennal tissues prepared by cryofixation is the smooth appearance of membrane structures (*Shanbhag et al., 1999*; *Shanbhag et al., 2000*; *Steinbrecht, 1980*; *Steinbrecht and Müller, 1987*). In the insect antenna, auxiliary cells extend

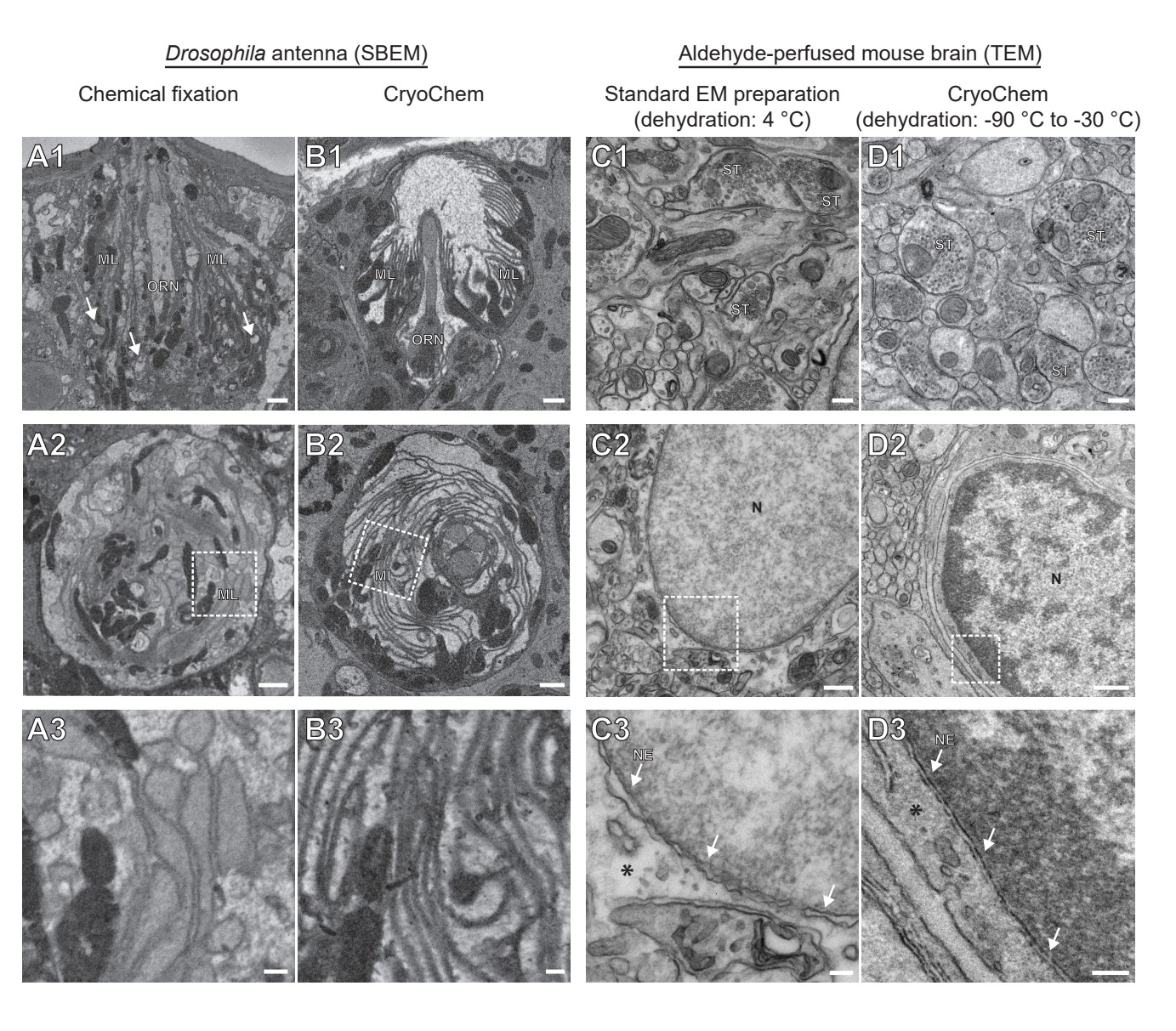

**Figure 2.** CryoChem method offers high-quality ultrastructural preservation and sufficient *en bloc* staining for SBEM. EM images were acquired to assess the morphology of CCM-processed tissues. (A-B) The quality of preservation was markedly improved in the CCM-processed *Drosophila* antenna compared to the chemically fixed counterpart. Pixel resolution of SBEM images (x,y): 6.5 nm. (A1 and B1) Unlike the CCM-processed antenna, the chemically fixed antenna showed signs of extraction (arrow) and disorganized membranes. ORN: olfactory receptor neuron; ML: microlamella. Scale bars: 1 μm. (A2 and B2) The microlamellae were well-preserved in the CCM-processed antenna, compared to the chemically fixed samples. Scale bars: 1 μm. (A3 and B3) In the enlarged views of the boxed regions, the microlamellae in the CCM-processed antenna appeared uniform in size and shape, unlike the chemically fixed ones which were distorted. Scale bars: 200 nm. (C-D) CCM enhanced the morphological preservation of aldehyde-perfused mouse brain. The initial dehydration in standard EM preparation took place on ice for 1 hr, but it occurred during freeze-substitution at −90 °C to −30 °C for over 5 days in CCM processing. (C1 and D1) The smoothness of membranes was improved by CCM processing. ST: synaptic terminal. Scale bars: 200 nm. Pixel resolution of TEM images (x,y): 1.92 nm. (C2 and D2) The preservation of nuclear envelope was improved by CCM processing. N: nucleus. Scale bars: 500 nm. Pixel resolution (x,y): 2.88 nm. (C3 and D3) In the enlarged views of the boxed regions, the nuclear envelope (NE; arrows) appeared smoother and the cytoplasmic density (asterisk) was increased with CCM processing. We note that the chromatin was more heavily stained in the CCM-processed specimen, likely due to the additional exposure to uranyl acetate during freeze-substitution. Scale bars: 100 nm. Pixel resolution (x, y): 1.14 nm.

DOI: https://doi.org/10.7554/eLife.35524.004

The following video and figure supplement are available for figure 2:

*Figure 2 continued*

**Figure supplement 1.** TEM images showed well-preserved ultrastructures in the CCM-processed HEK 293T cells.
DOI: https://doi.org/10.7554/eLife.35524.005

**Figure 2—video 1.** A SBEM volume from a CryoChem-processed *Drosophila* antenna.
DOI: https://doi.org/10.7554/eLife.35524.006

microlamellae to surround the olfactory receptor neurons (ORNs), forming the most membrane-rich regions in the antenna. We therefore focused on this structure to evaluate the quality of morphological preservation afforded by our method. In the CCM-processed antennal tissues, we found that the delicate structure of the microlamellae was well-preserved (*Figure 2B* and *Figure 2—video 1*), unlike the chemically fixed counterparts in which microlamellae were disorganized and distorted (*Figure 2A*) (*Steinbrecht, 1980*). Furthermore, there were numerous signs of extraction of cellular materials in the chemically fixed antenna (*Figure 2A1*, arrows), but not in the CCM-processed specimen (*Figure 2B*). Importantly, the overall ultrastructural preservation achieved through CCM resembles that obtained by standard cryofixation and freeze-substitution protocols (*Shanbhag et al., 1999*; *Shanbhag et al., 2000*). This observation also suggests that the rehydration step in CCM leads to little, if any, swelling in the antenna tissue.

In contrast to fly antennae, which can be dissected expeditiously and frozen in the live state, certain tissues (e.g., mouse brain) are difficult to cryofix from life without tissue damage caused by anoxia or mechanical stress associated with dissection. In these cases, cryofixation can be performed after aldehyde perfusion and still produce quality morphological preservation (*Sosinsky et al., 2008*). To test whether CCM can improve morphological preservation of aldehyde-perfused samples, we cryofixed vibratome sections (100 μm) from an aldehyde-perfused mouse brain and processed the sample with CCM. Compared to specimens processed by a standard EM preparation method that involved dehydration on ice (*Figure 2C*), the CCM-processed samples, which were initially dehydrated through freeze-substitution, showed smoother membranes and an increase in cytoplasmic density (*Figure 2D*). This result indicates an improvement in morphological preservation and agrees with our previous observation that cellular morphology can be markedly improved even when cryofixation is performed after aldehyde perfusion (*Sosinsky et al., 2008*).

Of note, we adopted a high-contrast *en bloc* staining protocol (*Deerinck et al., 2010*; *Tapia et al., 2012*; *West et al., 2010*; *Williams et al., 2011*) when processing the *Drosophila* antennae and mouse brain. An adequate level of heavy metals was incorporated into these cryofixed samples to allow for successful imaging by SBEM (*Figures 2B* and *3*), even without nitrogen gas injection to dissipate any charge build-up that often occurs on samples of low conductivity (*Deerinck et al., 2018*) (*Figure 2—video 1* and *Figure 3D*). This *en bloc* staining protocol is normally reserved only for chemically fixed tissues, but is now made compatible with cryofixed samples by CCM.

## CryoChem method enables DAB labeling in cryofixed samples expressing APEX2

Next, we determined if DAB labeling reaction can be performed in cryofixed samples with CCM (*Figure 3A*). Using the CCM-processed cultured cells expressing APEX2, we observed DAB labeling in the targeted organelle (mitochondria) in the transfected cells, compared to the untransfected controls (*Figure 3B*). We further validated this approach in a CCM-processed *Drosophila* antenna; successful DAB labeling was also detected in genetically identified ORNs expressing APEX2 with X-ray microscopy (*Figure 3—video 1*). This imaging technique facilitates the identification of the region of interest for SBEM (*Figure 3C*), as we and others reported previously (*Bushong et al., 2015*; *Ng et al., 2016*). Crucially, we demonstrated that an EM volume of a genetically labeled, cryofixed ORN can be acquired with SBEM, which allowed for an accurate 3D reconstruction of the ORN through semi-automated segmentation (*Figure 3D*). Taken together, these results demonstrate that CCM can reliably generate DAB labeling by genetically encoded EM tags in cryofixed samples.

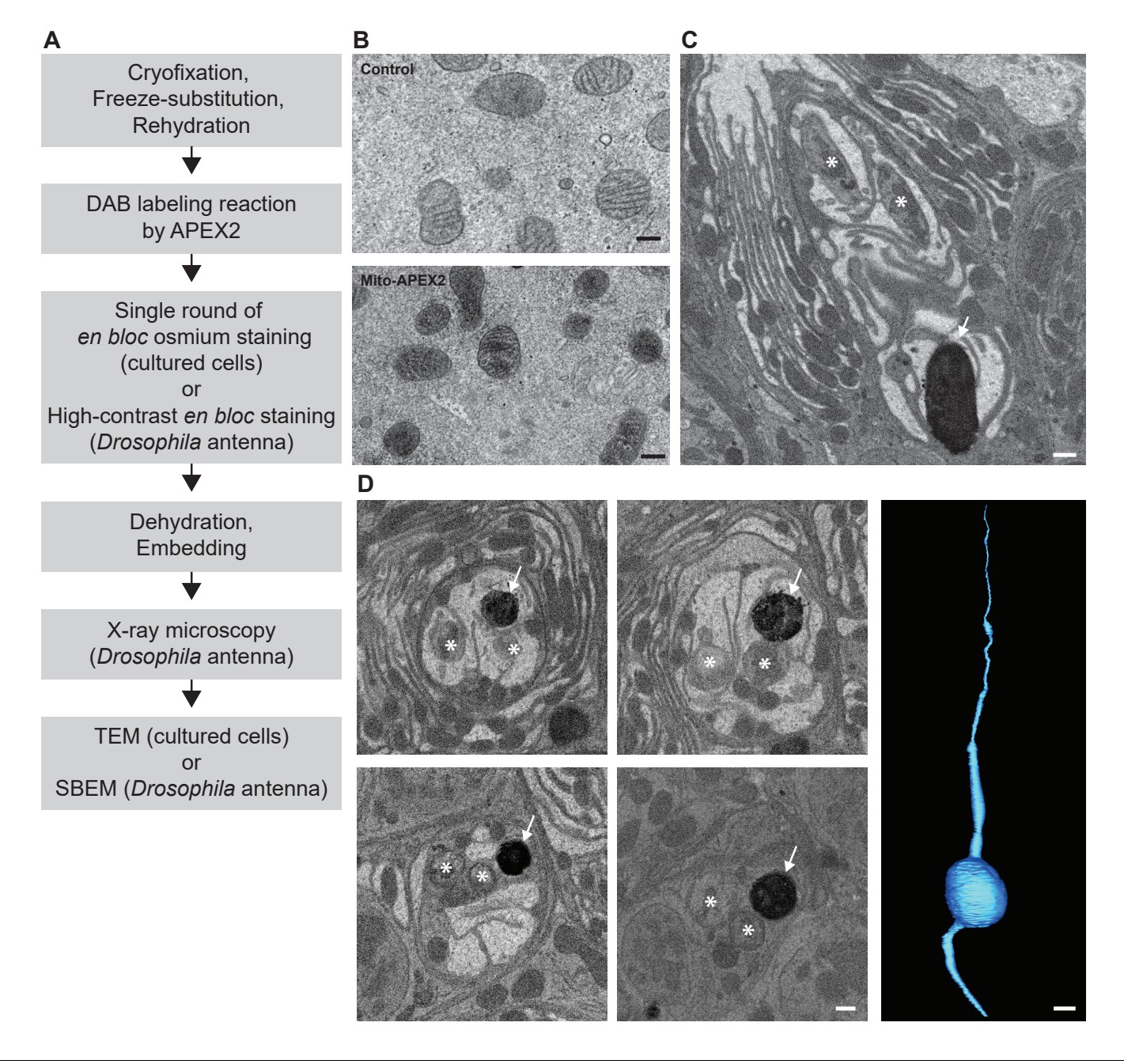

**Figure 3.** CryoChem method enables DAB labeling by APEX2 in cryofixed tissues. In the CCM-processed cultured cells and *Drosophila* antennae, DAB labeling was observed in cells expressing APEX2. (A) Flowchart for DAB labeling of target structures expressing APEX2 in CCM-processed samples. In our experiments, the cultured cells were imaged with TEM and the *Drosophila* antennae were imaged with X-ray microscopy, followed by SBEM. (B) Mitochondria in the HEK 293T cell transfected with Mito-APEX2 were labeled with DAB (bottom panel), in contrast to the untransfected control cell (top panel). Scale bars: 200 nm. Pixel resolution (x,y): 3.97 nm (top panel); 2.88 nm (bottom panel). (C) An APEX2-expressing olfactory receptor neuron (ORN) was labeled with DAB (arrow) in the *Drosophila* antenna (*10XUAS-myc-APEX2-Orco; Or47b-GAL4*). Asterisks denote ORNs without APEX2 expression. Scale bar: 500 nm. Pixel resolution of SBEM images (x,y): 6.5 nm. (D) A series of SBEM images showing the same DAB labeled *Drosophila* ORN (arrow) in different planes of section. Asterisks denote ORNs without APEX2 expression. The images were acquired using standard imaging methods without charge compensation by nitrogen gas injection (*Deerinck et al., 2018*). These images, together with the rest of the EM volume acquired using SBEM, enabled semi-automatic segmentation and 3D reconstruction of the labeled ORN (right panel). Scale bars: 500 nm for SBEM images, 2 μm for the 3D model of ORN. SBEM imaging parameters: Z step: 50 nm; Z dimension: 1200 sections; raster size: 12k × 9k; pixel size: 3.8 nm.

DOI: https://doi.org/10.7554/eLife.35524.007

The following video is available for figure 3:

*Figure 3 continued on next page*

*Figure 3 continued*
**Figure 3—video 1.** An X-ray micro-computed tomography volume from a CCM-processed *Drosophila* antenna showing DAB labeling in subsets of ORNs expressing APEX2 (*10XUAS-mCD8GFP-APEX2; Or22a-GAL4*).
DOI: https://doi.org/10.7554/eLife.35524.008

## Fluorescence is well-preserved in CryoChem-processed samples

To determine whether CCM is compatible with fluorescence microscopy, we first evaluated the degree to which fluorescence level is affected after CCM processing. Using confocal microscopy, we quantified GFP fluorescence in the soma of unfixed *Drosophila* ORNs and that from the CCM-processed samples after rehydration (*Figure 4*). Remarkably, GFP fluorescence intensities of the fresh and the CCM-processed ORNs were essentially indistinguishable with respect to their distributions (*Figure 4A*) and average levels (*Figure 4B*). This result indicates that CCM processing has little effect on GFP fluorescence in fly ORNs, likely due to the use of mild fixatives during freeze-substitution in our protocol. Similarly, we observed strong GFP signals in the mouse brain after the cryofixed sample was rehydrated (*Figure 4—figure supplement 1A*).

Next, we asked whether this observation also applies to another type of fluorescence protein. To this end, we examined tdTomato fluorescence in the mouse brain (*Figure 4—figure supplement 1B*). We note that tdTomato is not a variant of GFP and is instead derived from *Discosoma sp.*

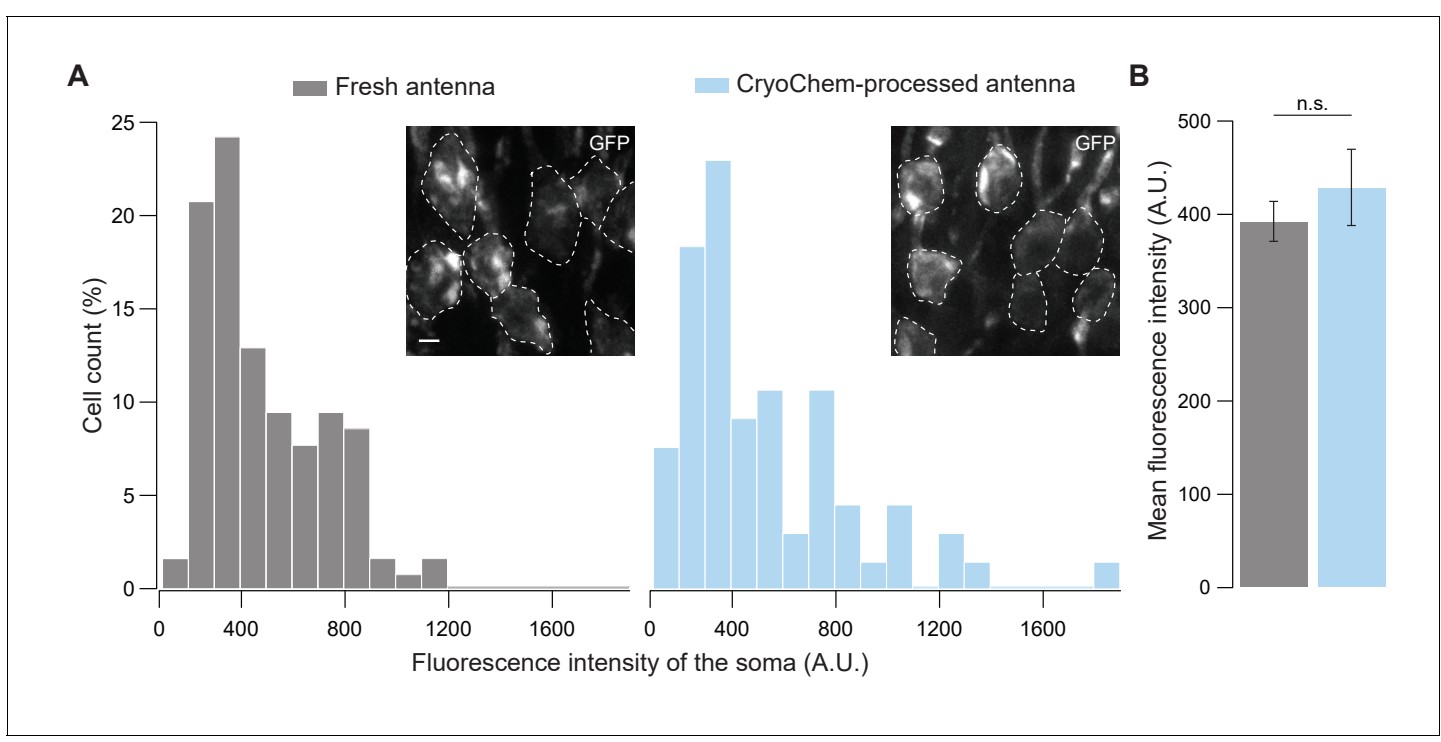

**Figure 4.** GFP fluorescence is well-preserved in CryoChem-processed samples. Confocal images were taken to quantify the level of GFP fluorescence in *Drosophila* ORNs. Antennae were collected from transgenic flies expressing GFP in a subset of ORNs. (**A**) GFP fluorescence intensity distributions of the ORN soma in the freshly-dissected, unfixed antennae (left panel) and the CCM-processed antennae (right panel) are not significantly different. p=0.810, Kolmogorov-Smirnov test. Insets show representative images, with ORN soma outlined. Scale bar: 2 μm. (**B**) Comparison of the average fluorescence intensities. GFP intensities are virtually identical between neurons in the unfixed antennae and the frozen-rehydrated antennae. n = 3 antennae, Error bars denote SEM, p=0.950, Mann-Whitney *U* Test.
DOI: https://doi.org/10.7554/eLife.35524.009

The following source data and figure supplement are available for figure 4:

**Source data 1.** GPF fluorescence intensities in unfixed and CryoChem-processed *Drosophila* ORN soma.
DOI: https://doi.org/10.7554/eLife.35524.010
**Figure supplement 1.** GFP and tdTomato fluorescence in a cryofixed-rehydrated mouse brain.
DOI: https://doi.org/10.7554/eLife.35524.011

fluorescence protein 'DsRed' (*Shaner et al., 2004*). Confocal images of the CCM-processed mouse brain showed that the tdTomato fluorescence was also well-preserved (*Figure 4—figure supplement 1B*) and we were able to detect the co-expression of GFP and tdTomato in a subpopulation of neurons (*Figure 4—figure supplement 1C*). Together, our results indicate that CCM-processed sample can serve as a robust substrate for fluorescence imaging. As such, CCM allows fluorescence imaging to be combined with DAB labeling and high-contrast *en bloc* staining in the same cryofixed sample, a critical advance to cryofixation-rehydration methods (*Dhonukshe et al., 2007*; *Ripper et al., 2008*; *Stierhof and El Kasmi, 2010*; *van Donselaar et al., 2007*).

## 3D correlative light and electron microscopy (CLEM) in CCM-processed samples expressing fluorescent markers

We took advantage of the fact that fluorescence microscopy can take place in a cryofixed sample before resin embedding to develop a protocol for 3D CLEM in CCM-processed specimens (*Figure 5A*, see Materials and methods for details), so that the correlation can be achieved in optimally preserved tissues. The protocol first uses the core CCM steps to deliver a frozen-rehydrated sample. Subsequently, DRAQ5 DNA stain is introduced to the sample to label the nuclei, which can then serve as fiducial markers for CLEM. Next, the region containing target cells expressing fluorescent markers is imaged with confocal microscopy, during which signals from DRAQ5 and fluorescent markers are both acquired. After confocal microscopy, the sample is *en bloc* stained with multiple layers of heavy metals (*Deerinck et al., 2010*; *Tapia et al., 2012*; *West et al., 2010*; *Williams et al., 2011*), then dehydrated and embedded as in a typical CCM protocol. Subsequently, the embedded sample is imaged with X-ray microscopy. The resulting micro-computed tomography volume can be registered to the confocal volume using the nuclei as fiducial markers, so that the region of interest (ROI) for SBEM can be identified. After SBEM imaging, the EM volume can be registered to the confocal volume in a similar fashion for 3D CLEM.

As a proof of principle, we performed 3D CLEM in an aldehyde-perfused, CCM-processed mouse brain expressing tdTomato in a subset of neurons. To this end, we first determined if DRAQ5 staining can be performed in a frozen-rehydrated specimen. Using confocal microscopy, we were able to observe DRAQ5 labeling of the nuclei in a cryofixed brain slice after rehydration (*Figure 5B*). We used the labeled nuclei as fiducial markers to register the X-ray volume with the confocal data (*Figure 5B*) and thereby target a ROI with tdTomato-expressing neurons for SBEM imaging.

Similarly, we were able to register the confocal volume to the SBEM volume (*Figure 5C*). Of note, the CLEM accuracy was ensured by using a subset of DRAQ5-labeled heterochromatin structures and their corresponding counterparts in EM as finer fiducial points (*Figure 5C*).

The fluorescent markers made it possible to identify the target cell bodies (*Figure 5D* and *Figure 5—video 1*) in the SBEM volume. The high accuracy of correlation achieved by our 3D CLEM protocol is demonstrated by the successful alignments of multiple ultrastructures: the fine neuronal processes (*Figure 5E*) and a subcellular heterochromatin structure that was not used as a fiducial marker (*Figure 5—figure supplement 1*). Lastly, we note that with CCM, fluorescence microscopy in cryofixed specimens takes place before *en bloc* EM staining. Therefore, our protocol does not require special resins for embedding and permits high-contrast staining with high concentrations of osmium tetroxide.

## Discussion

We described here a hybrid method, named CryoChem, which combines key advantages of cryofixation and chemical fixation to substantially broaden the applicability of the optimal fixation technique. With CCM, it is now possible to label target structures with DAB by a genetically encoded EM tag and deposit high-contrast *en bloc* staining in cryofixed tissues. In addition, with CCM, one can image cells expressing fluorescent markers before resin embedding and perform 3D CLEM in cryofixed specimens. Our method thereby provides an alternative to conventional cryofixation and chemical fixation methods.

The modular nature of CCM (*Figure 1*) makes it highly versatile as researchers can modify the modules to best suit their needs. For instance, to prevent over-staining, one can replace the high-contrast *en bloc* staining step (osmium-thiocarbohydrazide-osmium and uranyl acetate) (*Deerinck et al., 2010*; *Tapia et al., 2012*; *West et al., 2010*; *Williams et al., 2011*) with a single

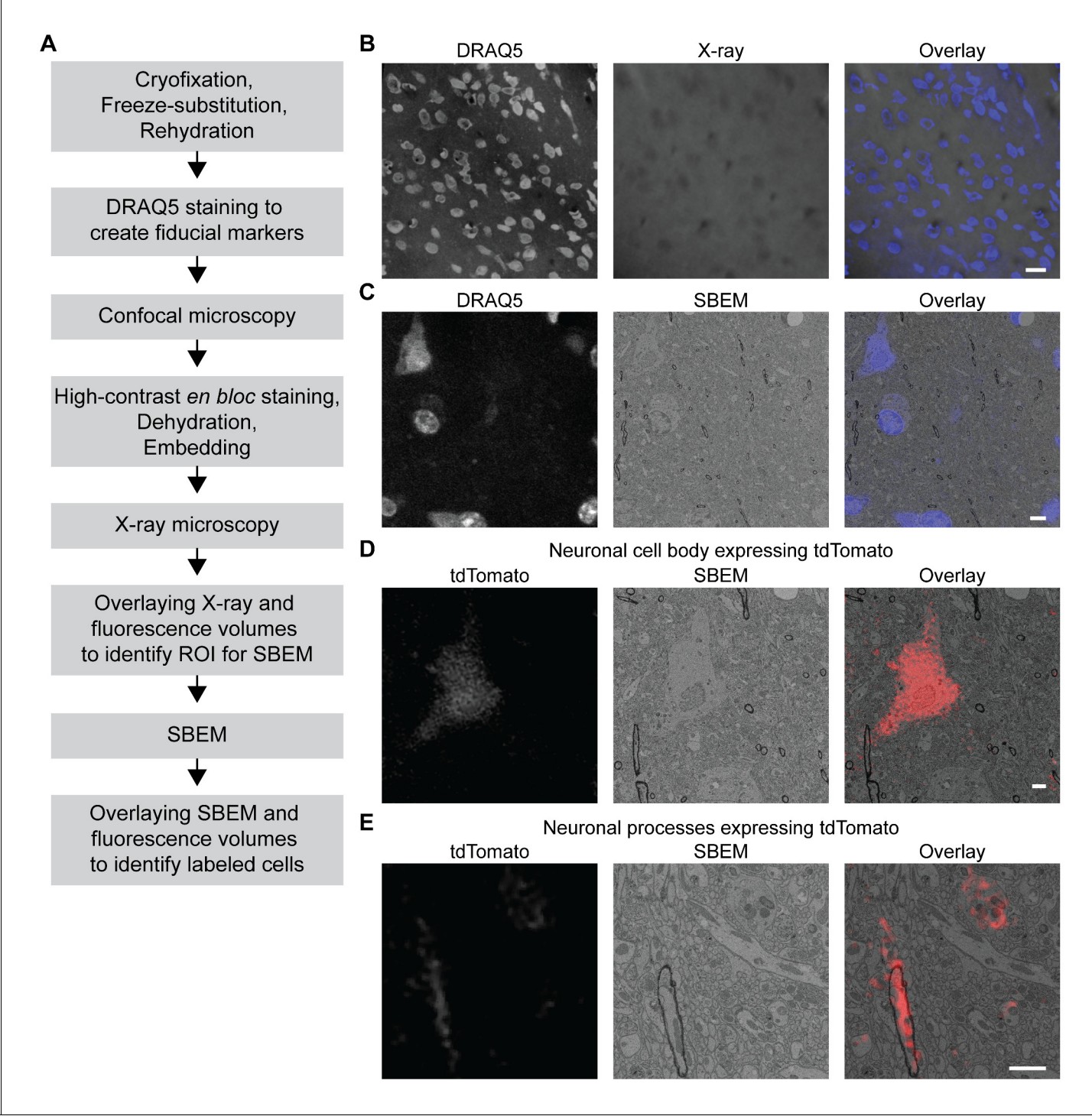

**Figure 5.** 3D correlated light and electron microscopy (CLEM) in CCM-processed mouse brain. Mouse brain slices with fluorescently labeled neurons were processed with CCM, imaged with confocal microscopy, X-ray microscopy and SBEM for 3D CLEM. (**A**) Flowchart for performing 3D CLEM with CCM-processed samples. Similar to a typical CCM protocol, cryofixed samples are first freeze-substituted and rehydrated. The frozen-rehydrated sample is then stained with DRAQ5 to label DNA in the nuclei. Next, the region of interest (ROI) is identified using confocal microscopy based on fluorescent signals, while the DRAQ5 signals are also acquired to serve as fiducial markers. Subsequently, the sample is stained, dehydrated and embedded for X-ray microscopy and SBEM. Using the DRAQ5 signals as fiducial markers, the confocal volumes can be registered to the X-ray volume such that the ROI for SBEM can be identified. Once the SBEM volume is acquired, it can be registered to the confocal volumes based on the positions of the nuclei for 3D CLEM. (**B**) An example of the DRAQ5 fluorescence signals (left), the corresponding ROI in X-ray volume (middle) and the overlay (right). This image registration process facilitates ROI identification in SBEM. Scale bar: 20 µm. (**C**) DRAQ5 fluorescence labeling served as fiducial

*Figure 5 continued on next page*

*Figure 5 continued*
points for registering the confocal volume to the SBEM volume. Scale bar: 5 μm. (**D**) The cell body of a tdTomato-expressing neuron (left) was identified in the SBEM volume (middle) through CLEM (right). (**E**) Neuronal processes expressing tdTomato (left) were also identified in the SBEM volume (middle) through CLEM (right). Scale bars: 2 μm, for both (**D**) and (**E**). SBEM imaging parameters: Z step: 70 nm; Z dimension: 695 sections; raster size: 10k × 15k; pixel size: 6.8 nm.
DOI: https://doi.org/10.7554/eLife.35524.012
The following video and figure supplement are available for figure 5:

**Figure supplement 1.** A correlation of subcellular structure can be achieved using 3D CLEM performed on the CCM-processed mouse brain.
DOI: https://doi.org/10.7554/eLife.35524.013
**Figure 5—video 1.** A volume showing 3D CLEM in the CCM-processed mouse brain.
DOI: https://doi.org/10.7554/eLife.35524.014

round of osmium tetroxide staining for thin section TEM (*Figures 2C, D* and *3B*) or electron tomography. In addition, CCM is essentially compatible with a wide range of reactions catalyzed by EM tags other than APEX2 (*Ellisman et al., 2015*). For example, the protein labeling reactions mediated by miniSOG (*Shu et al., 2011*) and the tetracysteine-based methods using FlAsH and ReAsH (*Gaietta et al., 2002*), or the non-protein biomolecule labeling reactions using Click-EM (*Ngo et al., 2016*) or ChromEM (*Ou et al., 2017*). The versatility of CCM will likely expand the breadth of biological questions that can be addressed using cryofixed samples.

In addition to using EM tags, we have also developed a 3D CLEM protocol (*Figure 5A*) that allows optimally preserved EM structures to be genetically labeled with fluorescent markers in CCM-processed tissues. In contrast to EM tags, fluorescent markers do not generate electron-dense products (e.g. DAB polymers) that can obscure the subcellular structures. Moreover, with multicolor CLEM, one can utilize multiple readily available genetically encoded fluorescent markers to label different target structures or cells. Using the 3D CLEM protocol, one could also pinpoint labeled subcellular structures (e.g., microtubules) or proteins (e.g., ion channels) in an EM volume with super-resolution microscopy. Furthermore, the ability to genetically label target neurons with fluorescent markers or EM tags in CCM-processed tissues can facilitate circuit reconstructions of identified neurons in optimally preserved specimens.

The advantages of CCM make it particularly suited for addressing biological questions that require optimal and rapid preservation of a genetically labeled structure. For example, to construct an accurate model to describe the biophysical properties of a neuron, it is essential to acquire morphological measurements based on faithfully preserved ultrastructures. CCM processing provides such an opportunity; we were able to obtain a 3D reconstruction of a genetically labeled *Drosophila* ORN at nanoscale resolution with quality morphological preservation (*Figure 3D*). In addition, by combining CCM with Flash-and-Freeze EM (*Watanabe et al., 2013a*, *2013b*, *2014a*, *2014b*) and electron tomography, it is possible to capture the fast morphological changes of genetically labeled vesicles in 3D during synaptic transmission.

Despite its versatility, multiple factors could potentially limit the applicability of CCM. First, given that the core fixation step of CCM is cryofixation, the size of the sample is constrained by the vitrification limit of up to approximately 500 μm (*Dahl and Staehelin, 1989*; *McDonald, 1999*; *Moor, 1987*; *Shimoni et al., 1998*). In addition, freeze damage due to ice crystal formation can occur (*Korogod et al., 2015*; *Ripper et al., 2008*; *Shanbhag et al., 2000*). Therefore, one should be mindful of freeze damage when performing ultrastructural analysis. Moreover, CCM can only improve the temporal resolution of biological events captured if the specimen is frozen in the live state, but not when the sample was first chemically fixed (e.g. aldehyde-perfused mouse brain). Finally, there are also concerns that some molecules may be lost during rehydration if they are not properly fixed during freeze-substitution (*Ripper et al., 2008*).

In conclusion, CCM is applicable to addressing questions in diverse tissue types, as demonstrated here with cultured mammalian cells or tissues of *Drosophila* antennae and mouse brains. Notably, identical solutions and experimental conditions were used for these different tissues in all core steps (*Figure 1*). Thus, the protocol described here can likely be readily adapted to cells and tissues of other biological systems. In addition, we demonstrated that CCM can further improve the ultrastructure of an aldehyde-perfused brain (*Figure 2C and D*). Given that aldehyde perfusion is often

required for the dissection of deeply embedded or fragile tissues, the compatibility of CCM with aldehyde fixation further broadens the applicability of the method.

# Materials and methods

## Key resources table

| Reagent type (species) or resource | Designation | Source or reference | Identifiers | Additional information |
|---|---|---|---|---|
| genetic reagent (*Drosophila melanogaster*) | Or47b-GAL4 | (*Fishilevich and Vosshall, 2005*) | RRID:BDSC_9984 | |
| genetic reagent (*D. melanogaster*) | Or22a-GAL4 | (*Dobritsa et al., 2003*) | RRID:BDSC_9951 | |
| genetic reagent (*D. melanogaster*) | 10XUAS-myc-APEX2-Orco | this study | | see Materials and methods |
| genetic reagent (*D. melanogaster*) | 10XUAS-mCD8GFP-APEX2 | this study | | see Materials and methods |
| genetic reagent (*Mus musculus*) | B6.Cg-Crht^m1(cre)Zjh/J | Jackson Laboratory | 031559 RRID:IMSR_JAX:031559 | |
| genetic reagent (*M. musculus*) | B6.Cg-Gt.ROSA.26 Sor^tm14(CAG-tdTomato)Hze/J | Jackson Laboratory | 007914 RRID:IMSR_JAX:007914 | |
| genetic reagent (*M. musculus*) | TH-GFP | (*Kessler et al., 2003*) | | |
| cell line | HEK 293T | ATCC | CRL-3216 RRID:CVCL_0063 | |
| antibody | DRAQ5 | Cell Signaling Technology | 4084 | 1:1000 |
| recombinant DNA reagent (plasmids) | pcDNA3-Mito-V5-APEX2 | (*Lam et al., 2015*) | Addgene_72480 | |
| recombinant DNA reagent (plasmids) | APEX2 DNA | (*Lam et al., 2015*) | Addgene_49386 | |
| recombinant DNA reagent (plasmids) | APEX2-Orco | this study | | see Materials and methods |
| recombinant DNA reagent (plasmids) | mCD8GFP-APEX2 | this study | | see Materials and methods |
| chemical compound, drug | paraformaldehyde | Fisher Scientific | 04042–500 | |
| chemical compound, drug | glutaraldehyde | Ted Pella | 18426 | |
| chemical compound, drug | sodium cacodylate | Ted Pella | 18851 | |
| chemical compound, drug | $CaCl_2$ | Sigma-Aldrich | 223506 | |
| chemical compound, drug | glycine | Bio-Rad Laboratories | 161–0718 | |
| chemical compound, drug | BSA | Sigma-Aldrich | 9048-46-8 | |
| chemical compound, drug | 1-hexadecene | Sigma-Aldrich | H2131 | |
| chemical compound, drug | uranyl acetate | Electron Microscopy Sciences | 22400 | |
| chemical compound, drug | methanol | Fisher Scientific | A412-4 | |
| chemical compound, drug | acetone | ACROS Organics | AC326800010 | |
| chemical compound, drug | HEPES | Gibco | 15-630-080 | |
| chemical compound, drug | diaminobenzidine; DAB | Sigma-Aldrich | D5637 | |
| chemical compound, drug | $H_2O_2$ | Fisher Scientific | H325-100 | |
| chemical compound, drug | osmium tetroxide; $OsO_4$ | Electron Microscopy Sciences | 19190 | |
| chemical compound, drug | potassium ferrocyanide | Mallinckrodt | 6932 | |
| chemical compound, drug | thiocarbohydrazide | Electron Microscopy Sciences | 21900 | |
| chemical compound, drug | Durcupan ACM resin component A | Sigma-Aldrich | 44611 | |
| chemical compound, drug | Durcupan ACM resin component B | Sigma-Aldrich | 44612 | |

*Continued on next page*

*Continued*

| Reagent type (species) or resource | Designation | Source or reference | Identifiers | Additional information |
|---|---|---|---|---|
| chemical compound, drug | Durcupan ACM resin component C | Sigma-Aldrich | 44613 | |
| chemical compound, drug | Durcupan ACM resin component D | Sigma-Aldrich | 44614 | |
| chemical compound, drug | conductive silver epoxy | Ted Pella | 16043 | |
| software, algorithm | IMOD | (*Kremer et al., 1996*) | RRID:SCR_003297 | http://bio3d.colorado.edu/imod/ |
| software, algorithm | ImageJ | NIH | RRID:SCR_003070 | https://imagej.nih.gov/ij/ |
| software, algorithm | Amira 6.3 | ThermoFisher | RRID:SCR_014305 | |
| other | Aclar | Electron Microscopy Sciences | 50426 | |
| other | FocusClear | Cedarlane Labs | FC-101 | |

## Cultured cells preparation

HEK 293T cells (ATCC, Gaithersburg, MD) were grown on 1.2 mm diameter punches of Aclar (two mil thick; Electron Microscopy Sciences, Hatfield, PA) for 48 hr, in a humidified cell culture incubator with 5% $CO_2$ at 37°C. Authentication was guaranteed by ATCC, including STR profiling. The cells were negative for mycoplasma, as confirmed by using the Universal Mycoplasma Detection Kit (ATCC, Gaithersburg, MD). The culture medium used was DMEM (Mediatech Inc., Manassas, VA) supplemented with 10% fetal bovine serum (Gemini Bio-Products, West Sacramento, CA). The cells were transfected with Lipofectamine 2000 (Invitrogen, Carlsbad, CA) with a plasmid carrying APEX2 targeted to mitochondria (pcDNA3-Mito-V5-APEX2, Addgene #72480; *Lam et al., 2015*). At 24 hr after transfection, the cells were used for CCM processing.

## DNA constructs and *Drosophila* transgenesis

Orco cDNA was a gift from Dr. Aidan Kiely, and APEX2 DNA was acquired from Addgene (APEX2-NES, #49386). Membrane targeting of APEX2 was achieved by fusing the marker protein to the C-terminus of mCD8GFP or to the N-terminus of Orco. Briefly, gel-purified PCR fragments of mCD8GFP, APEX2, and/or Orco were pieced together with Gibson Assembly following manufacturer's instructions (New England Biolabs, Ipswich, MA). A linker (SGGGG) was added between APEX2 and its respective fusion partner. In the APEX2-Orco construct, a myc tag was included in the primer and added to the N-terminus of APEX2 to enable the detection of the fusion protein by immunostaining. To facilitate Gateway Cloning (ThermoFisher Scientific, Waltham, MA), the attB1 and attB2 sites were included in the primers and added to the ends of the Gibson assembly product by PCR amplification. The PCR products were then purified and cloned into pDONR221 vectors via BP Clonase II (Life Technologies, Carlsbad, CA). The entry clones were recombined into the pBID-UASC-G destination vector (*Wang et al., 2012*) using LR Clonases II (Life Technologies, Carlsbad, CA).

*Drosophila* transgenic lines were derived from germline transformations using the ΦC31 integration systems (*Groth et al., 2004*; *Markstein et al., 2008*). All transgenes described in this study were inserted into the attP40 landing site on the second chromosome (BestGene Inc., Chino Hills, CA). Target expression of APEX2 and mCD8GFP in the ORNs was driven by the Or47b-GAL4 driver (#9984, Bloomington Drosophila Stock Center; *Fishilevich and Vosshall, 2005*; *Figures 2–4*) or the Or22a-GAL4 driver (#9951, Bloomington Drosophila Stock Center; *Dobritsa et al., 2003*; *Figure 3—video 1*). Flies were raised on standard cornmeal food at 25°C in a 12:12 light-dark cycle.

## *Drosophila* antennae preparation

Six to eight days old flies were cold anesthetized and then pinned to a Sylgard dish. The third segments of the antennae were removed from the head of the fly with a pair of fine forceps and then immediately transferred to a drop of 1X PBS on the dish. With a sharp glass microelectrode, a hole was poked in the antenna to facilitate solution exchange. It is critical that the tissue remained in PBS at all times to prevent deflation. The antenna should remain plump and maintain its shape prior to cryofixation.

## Chemical fixation of *Drosophila* antennae

Antennae were dissected as described above, and then incubated at 4°C for 18 hr in Karnovsky fixatives: 2% paraformaldehyde (Fisher Scientific, Hampton, NH)/2.5% glutaraldehyde (Ted Pella, Redding, CA)/2 mM CaCl$_2$ (Sigma-Aldrich, St. Louis, MO) in 0.1 M sodium cacodylate (Ted Pella, Redding, CA). Next, samples were washed in 0.1 M sodium cacodylate for 10 min and in a solution of 100 mM glycine (Bio-Rad Laboratories, Hercules, CA) in 0.1 M sodium cacodylate for another 10 min, and twice more in 0.1 M sodium cacodylate. All washing steps were performed on ice. The following *en bloc* heavy metal staining, dehydration and resin embedding steps were carried out as described in the CryoChem Method section below.

## Transgenic mice and virus-mediated gene transfer

Animals were handled in accordance with the guidelines established by the *Guide for Care and Use of Laboratory Animals* and approved by UCSD Animal Care and Use Committee. To introduce GFP and tdTomato fluorescent markers in a mouse brain (*Figure 4—figure supplement 1*), GFP was expressed in the tyrosine hydroxylase (TH)-expressing neurons and tdTomato in the corticotropin releasing factor (CRF)-expressing neurons. A CRF driver mouse line (B6.Cg-Crh$^{tm1(cre)Zjh}$/J, Jackson laboratory) expressing CRE recombinase under the control of the Crh promoter/enhancer elements was first crossed to a tdTomato reporter line (B6.Cg-Gt.ROSA.26Sor$^{tm14(CAG-tdTomato)Hze}$/J, Jackson Laboratory). The progeny was then crossed to a TH-GFP mouse line (*Kessler et al., 2003*), obtaining a transgenic mouse stably expressing GFP in dopaminergic (TH$^+$) neurons and CRE/tdTomato in CRF-releasing neurons. To test the 3D CLEM protocol and the morphological preservation offered by CCM (*Figures 2C, D* and *5*, *Figure 5—figure supplement 1* and *Figure 5—video 1*), a similar strategy was used to genereate a mouse expressing CRE/tdTomato in CRF-releasing neurons.

## Mouse brain preparation

Mice were anesthetized with ketamine/xylazine and then transcardially perfused with Ringer's solution followed by 0.15 M sodium cacodylate containing 4% paraformaldehyde/0.2% (*Figure 4—figure supplement 1*) or 0.5% (*Figures 2C, D* and *5*, *Figure 5—figure supplement 1* and *Figure 5—video 1*) glutaraldehyde/2 mM CaCl$_2$. The animal was perfused for 10 min with the fixatives, and then the brain was removed and placed in ice-cold fixative for 1 hr. The brain was then cut into 100 μm thick slices using a vibrating microtome. Slices were either processed for chemical fixation (*Figure 2C*) or stored in ice-cold 0.15 M sodium cacodylate for around 4 hr until used for high-pressure freezing (*Figure 2D*, *Figure 4—figure supplement 1*, *Figure 5*, *Figure 5—figure supplement 1* and *Figure 5—video 1*).

## Chemical fixation of mouse brain

The aldehyde-perfused mouse brain slices were post-fixed in 2.5% glutaraldehyde for 20 min, then washed with 0.15 M sodium cacodylate five times for 5 min on ice. Next, the samples were incubated in 0.15 M sodium cacodylate with 100 mM glycine for 5 min on ice, then washed in 0.15 M sodium cacodylate similarly. The following *en bloc* heavy metal staining, dehydration and resin embedding steps were carried out as described in the CryoChem Method section below.

## CryoChem Method

### (I) Cryofixation by high-pressure freezing

### Cultured cells

Aclar disks were placed within the well of a 100 μm-deep membrane carrier. The cells were covered with the culture medium and then high-pressure frozen with a Leica EM PACT2 unit.

### *Drosophila* antennae

The third antennal segment was dissected as described above. Antennae from the same fly were transferred into the 100 μm-deep well of a type A planchette filled with 20% BSA (Sigma-Aldrich, St. Louis, MO) in 0.15 M sodium cacodylate. The well of the type A planchette was then covered with the flat side of a type B planchette to secure the sample. The samples were immediately loaded into a freezing holder and frozen with a high-pressure freezing machine (Bal-Tec HPM 010). Planchettes used for cryofixation were pre-coated with 1-hexadecene (Sigma-Aldrich, St. Louis, MO) to prevent

planchettes A and B from adhering to each other, so as to allow solution to reach the samples during freeze-substitution.

## Mouse brain slices

A 1.2 mm tissue puncher was used to cut a portion of hypothalamus expressing tdTomato (*Figure 2D*, *Figure 4—figure supplement 1*, *Figure 5*, *Figure 5—figure supplement 1* and *Figure 5—video 1*) and GFP (*Figure 4—figure supplement 1*) from a tissue slice. The tissue punch was placed into a 100 μm-deep membrane carrier and surrounded with 20% BSA in 0.15 M sodium cacodylate. The specimen was high-pressure frozen as described for the *Drosophila* antennae.

All frozen samples were stored in liquid nitrogen until further processing.

## (II) Freeze-substitution

Frozen samples in planchettes were transferred in a liquid nitrogen bath to cryo-vials containing the freeze-substitution solution. To prepare the freeze-substitution solution of 0.2% glutaraldehyde, 0.1% uranyl acetate, 2% methanol and 1% water in acetone, a 10 mL solution was prepared by adding 80 μL of 25% aqueous glutaraldehyde, 200 μL of 5% uranyl acetate (Electron Microscopy Sciences, Hatfield, PA) dissolved in methanol, and 20 μL of water to acetone (ACROS Organics, USA). Next, the sample vials were transferred to a freeze-substitution device (Leica EM AFS2) at −90 °C for 58 hr, from −90 °C to −60 °C for 15 hr (with the temperature raised at 2 °C/hr), at −60 °C for 15 hr, from −60 °C to −30 °C for 15 hr (at +2 °C/hr), and then at −30 °C for 15 hr. In the last hour at −30 °C, samples were washed three times in an acetone solution with 0.2% glutaraldehyde and 1% water for 20 min. The cryo-tubes containing the last wash were then transferred on ice for an hour.

## (III) Rehydration

The freeze-substituted samples were then rehydrated gradually in a series of nine rehydration solutions (see below). The samples were transferred from the freeze-substitution solution to the first rehydration solution (5% water, 0.2% glutaraldehyde in acetone) on ice for 10 min. The rehydration step was repeated in a stepwise manner until the samples were fully rehydrated in the final rehydration solution (0.1 M and 0.15 M sodium cacodylate for cells and antennae or mouse brain slices, respectively) (*van Donselaar et al., 2007*):

1. 5% water, 0.2% glutaraldehyde in acetone
2. 10% water, 0.2% glutaraldehyde in acetone
3. 20% water, 0.2% glutaraldehyde in acetone
4. 30% water, 0.2% glutaraldehyde in acetone
5. 50% 0.1 M HEPES (Gibco, Taiwan), 0.2% glutaraldehyde in acetone
6. 70%, 0.1 M HEPES, 0.2% glutaraldehyde in acetone
7. 0.1 M HEPES
8. 0.1 M / 0.15 M sodium cacodylate with 100 mM glycine
9. 0.1 M / 0.15 M sodium cacodylate

After rehydration, samples were removed from the planchettes using a pair of forceps under a stereo microscope to a 0.1 M (cells and antenna)/0.15 M (brain) sodium cacodylate solution in a scintillation vial on ice. It is important that subsequent DAB labeling and *en bloc* heavy metal staining are carried out in scintillation vials instead of the planchettes because metal planchettes may react with the labeling or staining reagents.

## (IV) DRAQ5 staining

Mouse brain slices were incubated in DRAQ5 (1:1000 in 0.15 M sodium cacodylate buffer; Cell Signaling Technology, Danvers, MA) on ice for 60 min. Then the samples were washed in 0.15 M sodium cacodylate three times for 10 min on ice before fluorescence imaging.

## (V) Fluorescence imaging

### *Drosophila* antennae

Freshly dissected or cryofixed-rehydrated antennae (*10XUAS-mCD8GFP-APEX2; Or47b-GAL4*) were mounted in FocusClear (Cedarlane Labs, Burlington, Canada) between two cover glasses (#1.5 thickness, 22 mm x 22 mm, Fisher Scientific, Hampton, NH) separated by two layers of spacer rings.

Confocal images were collected on an Olympus FluoView 1000 confocal microscope with a 60X water-immersion objective lens. The 488 nm laser was used to excite GFP and all images were acquired at the same laser power and gain to enable comparison between the fresh vs cryofixed-rehydrated samples.

## Mouse brain slices

After freeze-substitution and rehydration, the specimens were placed in ice-cold 0.15 M sodium cacodylate for imaging. Confocal images of GFP and tdTomato signals (*Figure 4—figure supplement 1*) were collected on a Leica SPE II confocal microscope with a 20X water-immersion objective lens using 488 nm and 561 nm excitation. Confocal volumes of DRAQ5 and tdTomato signals (*Figure 5*, *Figure 5—figure supplement 1*, *Figure 5—video 1*) were collected on an Olympus FluoView 1000 confocal microscope with a 20X air and 60X water objectives using 561 nm and 633 nm excitation.

## (VI) DAB labeling of target structures by APEX2
### Cultured cells

Samples were transferred to a 0.05% DAB (Sigma-Aldrich, St. Louis, MO) solution in 0.1 M sodium cacodylate for 5 min on ice to allow DAB to diffuse into the tissue. To label the mitochondria in the APEX2-expressing cells, samples were then transferred to a 0.05% DAB solution with 0.015% $H_2O_2$ (Fisher Scientific, Hampton, NH) in 0.1 M sodium cacodylate until DAB labeling was visible under a microscope (~5 min on ice). After the reaction, samples were washed three times with 0.1 M sodium cacodylate on ice for 10 min.

### *Drosophila* antennae

Samples were first placed into a 0.05% DAB solution in 0.1 M sodium cacodylate for an hour on ice to allow DAB to access target neurons underneath the cuticle in the antenna. To label APEX2-expressing ORNs, antennae were then transferred into a 0.05% DAB solution with 0.015% $H_2O_2$ in 0.1 M sodium cacodylate for an hour on ice. After the reaction, samples were washed three times with 0.1 M sodium cacodylate on ice for 10 min.

## (VII) *en bloc* heavy metal staining for TEM and SBEM

For TEM: Cultured cells and mouse brain slices were incubated in 2% $OsO_4$ (Electron Microscopy Sciences, Hatfield, PA)/1.5% potassium ferrocyanide (Mallinckrodt, Staines-Upon-Thames, UK)/2 mM $CaCl_2$ in 0.1 M (cells) or 0.15 M (brain) sodium cacodylate for an hour on ice. Then samples were washed in water five times for 5 min on ice prior to the dehydration step detailed below.

For SBEM: *Drosophila* antennae and mouse brain slices were incubated in 2% $OsO_4$/1.5% potassium ferrocyanide/2 mM $CaCl_2$ in 0.1 M (antennae) or 0.15 M (brain) sodium cacodylate for an hour at room temperature. Then samples were washed in water five times for 5 min and transferred to 0.5% thiocarbohydrazide (filtered with 0.22 μm filter before use; Electron Microscopy Sciences, Hatfield, PA) for 30 min at room temperature. Samples were washed in water similarly and incubated in 2% $OsO_4$ for 30 min at room temperature. Afterwards, samples were rinsed with water, then transferred to 2% aqueous uranyl acetate (filtered with 0.22 μm filter) at 4 °C overnight. In the next morning, samples were first washed in water five times for 5 min and then subjected to the dehydration steps detailed below.

## (VIII) Dehydration

Samples were dehydrated with a series of ethanol solutions and acetone in six steps of 10 min each: 70% ethanol, 90% ethanol, 100% ethanol, 100% ethanol, 100% acetone, 100% acetone. All ethanol dehydration steps were carried out on ice, and the acetone steps at room temperature. The first acetone dehydration step was carried out with ice-cold acetone, and the second one was with acetone kept at room temperature.

## (IX) Resin infiltration

### Cultured cells

Samples were transferred to a Durcupan ACM resin/acetone (1:1) solution for an hour on a shaker at room temperature. The samples were then transferred to fresh 100% Durcupan ACM resin overnight and subsequently placed in fresh resin for four hours. While in 100% resin, samples were placed in a vacuum chamber on a rocker to facilitate the removal of residual acetone. Finally, the samples were embedded in fresh resin at 60 °C for two days.

### *Drosophila* antennae and mouse brain slices

Samples were transferred to a Durcupan ACM resin/acetone (1:1) solution overnight on a shaker. The next day, samples were transferred into fresh 100% Durcupan ACM resin twice, with six to seven hours apart. While in 100% resin, samples were placed in a vacuum chamber on a rocker to facilitate the removal of residual acetone. After the overnight incubation in 100% resin, samples were embedded in fresh resin at 60 °C for at least two days.

Durcupan ACM resin (Sigma-Aldrich, St. Louis, MO) composition was 11.4 g component A, 10 g component B, 0.3 g component C, and 0.1 g component D.

## X-ray microscopy (microcomputed tomography)

### *Drosophila* antennae

Microcomputed tomography (microCT) was performed on resin-embedded specimens using a Versa 510 X-ray microscope (Zeiss). Flat-embedded specimens were glued to the end of an aluminum rod using cyanoacrylic glue. Imaging was performed with a 40X objective using a tube current of 40 kV and no source filter. Raw data consisted of 1601 projection images collected as the specimen was rotated 360 degrees. The voxel dimension of the final tomographic reconstruction was 0.4123 μm.

### Mouse brain slices

X-ray microscopy scan was collected of a resin-embedded sample at 80 kVp with a voxel size of 0.664 μm prior to mounting for SBEM imaging. A second scan was collected of the mounted specimen at 80 kVp with 0.7894 μm voxels.

## Transmission electron microscopy

Ultrathin sections (70 nm) were collected on 300 mesh copper grids. Samples were post-stained with either Sato's lead solution only (cultured cells) or with 2% uranyl acetate and Sato's lead solution (mouse brain slices). Sections were imaged on an FEI Spirit TEM at 80 kV equipped with a 2k × 2k Tietz CCD camera.

## Serial Block-face scanning electron microscopy

### *Drosophila* antennae

Following microcomputed tomography to confirm proper orientation of region of interest, specimens were mounted on aluminum pins with conductive silver epoxy (Ted Pella, Redding, CA). The specimens were trimmed to remove excess resin above ROI and to remove silver epoxy from sides of specimen. The specimens were sputter coated with gold-palladium and then imaged using a Gemini scanning electron microscope (Zeiss) equipped with a 3View2XP and OnPoint backscatter detector (Gatan). Images were acquired at 2.5 kV accelerating voltage with a 30 μm condenser aperture and 1 μsec dwell time; Z step size was 50 nm; raster size was 12k × 9k and Z dimension was 1200 sections. Volumes were either collected in variable pressure mode with a chamber pressure of 30 Pa and a pixel size of 3.8 nm (*Figure 2—video 1* and *Figure 3D*) or using local gas injection (*Deerinck et al., 2018*) set to 85% and a pixel size of 6.5 nm (*Figures 2A, B* and *3C*). Volumes were aligned using cross correlation, segmented, and visualized using IMOD (*Kremer et al., 1996*).

### Mouse brain slices

SBEM was performed on a Merlin scanning electron microscope (Zeiss) equipped with a 3View2XP and OnPoint backscatter detector (Gatan). The volume was collected at 2 kV, with 6.8 nm pixels and

70 nm Z steps. Local gas injection (*Deerinck et al., 2018*) was set to 15% during imaging. The raster size was 10k × 15k and the Z dimension was 659 sections.

## Semi-automated segmentation of DAB-labeled *Drosophila* olfactory receptor neuron

The DAB-labeled *Drosophila* ORN was segmented in a semi-automated fashion using the IMOD software to generate the 3D model. The IMOD command line 'imodauto' was used for the auto-segmentation by setting thresholds to isolate the labeled cellular structures of interest. Further information about the utilities of 'imodauto' can be found in the IMOD manual (http://bio3d.colorado.edu/imod/doc/man/imodauto.html). Auto-segmentation was followed by manual proofreading and reconstruction by two independent proofreaders. The proofreaders used elementary operations in IMOD, most commonly the 'drawing tools' to correct the contours generated by 'imodauto'. Where 'imodauto' failed to be applied successfully, the proofreaders also used the 'drawing tools' to directly trace the outline of the labeled structure. The contours of ORNs generally do not vary markedly between adjacent sections. Therefore, alternate sections were traced for the reconstruction of some parts of the ORN dendrite.

## Quantification of fluorescence intensity

To quantify GFP fluorescence intensity shown in *Figure 4*, maximum intensity Z-projections were generated using ImageJ (NIH). Average fluorescence intensity in the background was subtracted from the fluorescence intensity of each cell body measured. Only non-overlapping cell bodies were quantified. Kolmogorov-Smirnov Test was performed on http://www.physics.csbsju.edu/stats/KS-test.html and Mann-Whitney *U* Test was performed using SigmaPlot 13.0 (Systat Software, San Jose, CA).

## Light and electron microscopy volume registration

To target tdTomato-expressing cells in the mouse brain for SBEM imaging, the confocal volumes collected in the frozen-rehydrated specimen was registered with the microCT volume of the resin-embedded sample, using a software tool developed in our lab. The resin-embedded specimen was then mounted and trimmed for SBEM based on the microCT volume. A second microCT scan of the mounted specimen allowed for precise targeting of the cells of interest with the Gatan stage for SBEM. After the SBEM volume was collected, the confocal and SBEM volumes were registered using the landmark tool of Amira 6.3 (ThermoFisher, Waltham, MA). Heterochromatin structures revealed by DRAQ5 labeling and visible in the SBEM volume were used as landmark points for the registration.

## Acknowledgements

We thank Aiden Keily for providing the Orco cDNA construct and R. Alexander Steinbrecht for advice on electron microscopy of *Drosophila* antennae. We also thank Edie Zhang and Martin Orden for assistance in segmentation of *Drosophila* olfactory receptor neuron. We also thank Andrea Thor and Mason Mackey for help with EM sample preparation and imaging. We also thank Steven Wasserman for comments on the manuscript. This work was supported by Frontiers of Innovation Scholars Program and Croucher Foundation Scholarship to TKT, the Ray Thomas Edwards Foundation Career Development Award, Kavli Institute for Brain and Mind Innovative Research Grant (2015–004) and NIH R01DC015519 to CYS. This work was also supported by a grant to MHE P41GM103412 from the National Institute of General Medical Sciences for support of the National Center for Microscopy and Imaging Research (NCMIR) technologies and instrumentation, the NIH R01GM086197 to DB and Kavli Institute for Brain and Mind Innovative Research Grant (2016–038) to DB and DD. The authors declare no conflicts of interest.

## Additional information

### Funding

| Funder | Grant reference number | Author |
|---|---|---|
| National Institute on Deafness and Other Communication Disorders | R01DC015519 | Chih-Ying Su |
| National Institute of General Medical Sciences | P41GM103412 | Mark H Ellisman |
| Croucher Foundation | | Tin Ki Tsang |
| Kavli Foundation | 2015-004 | Chih-Ying Su<br>Mark H Ellisman |
| Ray Thomas Edwards Foundation | | Chih-Ying Su |
| University of California, San Diego | Frontiers of Innovation Scholars Program | Tin Ki Tsang |
| National Institute of General Medical Sciences | R01GM086197 | Daniela Boassa |
| Kavli Foundation | 2016-038 | Daniela Boassa<br>Davide Dulcis |

The funders had no role in study design, data collection and interpretation, or the decision to submit the work for publication.

### Author contributions

Tin Ki Tsang, Conceptualization, Formal analysis, Funding acquisition, Validation, Investigation, Visualization, Methodology, Writing—original draft, Project administration, Writing—review and editing; Eric A Bushong, Conceptualization, Formal analysis, Validation, Investigation, Visualization, Methodology, Writing—original draft, Project administration, Writing—review and editing; Daniela Boassa, Resources, Formal analysis, Funding acquisition, Validation, Investigation, Writing—review and editing; Junru Hu, Investigation, Writing—review and editing; Benedetto Romoli, Resources, Investigation, Writing—review and editing; Sebastien Phan, Software, Visualization, Writing—review and editing; Davide Dulcis, Resources, Funding acquisition, Writing—review and editing; Chih-Ying Su, Conceptualization, Resources, Supervision, Funding acquisition, Visualization, Writing—original draft, Project administration, Writing—review and editing; Mark H Ellisman, Conceptualization, Resources, Supervision, Funding acquisition, Writing—original draft, Project administration, Writing—review and editing

### Author ORCIDs

Tin Ki Tsang (iD) http://orcid.org/0000-0003-1002-106X
Eric A Bushong (iD) http://orcid.org/0000-0001-6195-2433
Chih-Ying Su (iD) http://orcid.org/0000-0002-0005-1890

### Ethics

Animal experimentation: This study was performed in strict accordance with the recommendations in the Guide for the Care and Use of Laboratory Animals of the National Institutes of Health. All of the animals were handled according to approved institutional animal care and use committee (IACUC) protocols (#S15013 and S06211) of the University of California, San Diego. All mouse surgeries were performed under ketamine/xylazine anesthesia, and every effort was made to minimize suffering.

### Decision letter and Author response

Decision letter https://doi.org/10.7554/eLife.35524.027
Author response https://doi.org/10.7554/eLife.35524.028

## Additional files

### Supplementary files

• Transparent reporting form
DOI: https://doi.org/10.7554/eLife.35524.015

### Data availability

A source data file has been provided for Figure 4 (Figure 4-source data 1). The SBEM volume of a *Drosophila* antenna presented in this study has been deposited to the Cell Image Library. The SBEM volume, the tdTomato confocal volume and the DRAQ5 confocal volume used for 3D CLEM in a mouse brain (corresponding to Figure 5) have also been deposited to the Cell Image Library. The video of 3D CLEM in a mouse brain expressing tdTomato that corresponds to Figure 5-video supplement 1 has been deposited to the Cell Image Library (http://cellimagelibrary.org/groups/50451).

The following datasets were generated:

| Author(s) | Year | Dataset title | Dataset URL | Database, license, and accessibility information |
|---|---|---|---|---|
| Tsang TK, Bushong EA, Boassa D, Hu J, Romoli B, Phan S, Dulcis D, Su C-Y, Ellisman MH | 2018 | SBEM of *Drosophila* antenna (10XUAS-myc-APEX2-Orco; Or47b-GAL4) | http://cellimagelibrary.org/images/50452 | Publicly available at the Cell Image Library under accession number CIL:50452 |
| Tsang TK, Bushong EA, Boassa D, Hu J, Romoli B, Phan S, Dulcis D, Su C-Y, Ellisman MH | 2018 | SBEM volume used for 3D CLEM in mouse brain | http://cellimagelibrary.org/images/50451 | Publicly available at the Cell Image Library under accession number CIL:50451 |
| Tsang TK, Bushong EA, Boassa D, Hu J, Romoli B, Phan S, Dulcis D, Su C-Y, Ellisman MH | 2018 | tdTomato confocal volume used for 3D CLEM in a mouse brain | http://cellimagelibrary.org/images/50453 | Publicly available at the Cell Image Library under accession number CIL:50453 |
| Tsang TK, Bushong EA, Boassa D, Hu J, Romoli B, Phan S, Dulcis D, Su C-Y, Ellisman MH | 2018 | DRAQ5 confocal volume used for 3D CLEM in a mouse brain | http://cellimagelibrary.org/images/50454 | Publicly available at the Cell Image Library under accession number CIL:50454 |
| Tsang TK, Bushong EA, Boassa D, Hu J, Romoli B, Phan S, Dulcis D, Su C-Y, Ellisman MH | 2018 | Video of 3D CLEM in a mouse brain expressing tdTomato | http://cellimagelibrary.org/images/50401 | Publicly available at the Cell Image Library under accession number CIL:50401 |

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
