## [Decision Letter]

Thank you for submitting your article "High-quality ultrastructural preservation using cryofixation for 3D electron microscopy of genetically labeled tissues" for consideration by *eLife*. Your article has been reviewed by 3 peer reviewers, and the evaluation has been overseen by a Reviewing Editor, who served as one of the reviewers, and Eve Marder as the Senior Editor. The following individual involved in review of your submission has agreed to reveal his identity: Richard Leapman (Reviewer #3).

The reviewers have discussed the reviews with one another and the Reviewing Editor has drafted this decision to help you prepare a revised submission.

The manuscript reports a modified approach for sample fixation and staining that opens the possibility to combine cryofixation with 3D electron microscopic imaging and correlated light-electron microscopic analysis.

While the reviewers see the merits of this work, they raised a set of key concerns, and after discussion decided to request the following points for the revision of this manuscript:

- Proper citation and discussion of the literature as indicated by reviewer 2; clarification of which additional advances were made compared to the published protocols.

- More extensive discussion of the limitations of the approach (see reviewers 1 and 3): tissue volume effects, potentially problematic effects of the rehydration step; clarification for which examples the "improved temporal resolution" apply.

- Better description of the reported applications (see comments by reviewers 1 and 3 below): which resolution of correlated imaging was achieved; how does this depend on the specimen and procedure; how does it combine with circuit reconstruction (if this is envisioned to be possible in this approach).

All other points raised by the reviewers can be treated as recommendations:

*Reviewer #1:*

The manuscript "High-quality ultrastructural preservation using cryofixation for 3D electron microscopy of genetically labeled tissues" by Tsang et al. describes a novel fixation protocol providing the possibility to cryofix tissue followed by high-contrast heavy metal staining amenable to 3D electron microscopy and offering the possibility to perform correlated light EM imaging.

The challenges addressed by this manuscript are substantial and the solutions presented appear convincing. This is an important methodological contribution that in principle would merit a publication in *eLife*. However, I have a few concerns that in my view need to be addressed:

1) The concrete applications of the improved protocol remain a little unclear. Especially it is not clear at what resolution and for which volumes correlated LM-EM imaging becomes possible with this approach. The authors show data that speaks for cellular level correlation, i.e. the identification of stained cell bodies which has been possible before (Bock et al.; Briggman et al.) and would not be a major improvement. A correlation at the level of subcellular structures while still being able to acquire large 3D EM volumes would be highly commendable but it is not clear whether that is in fact achieved. Moreover, numbers on dataset size, resolution, etc. are missing from the Results section. They can be found in the Materials and methods but for such a methodological paper it would be absolutely critical in this reviewer’s view to have the numbers on acquired 3D EM volumes, resolution etc. in the main text. It appears that the SBEM volume acquired has a z extent of about 50 micrometers, however at a section thickness of 70 nanometers. 70 nanometers of section thickness would not routinely allow for neuronal circuitry construction – neither in fly nor in mammalian brains. This should be addressed or clarified. Again, all of these points would profit from a clearer description of the concrete application examples.

2) The description is not extremely clear, for instance the usage of the word "temporal" in the Abstract comes as a surprise and is only understandable after realizing that the authors are referring to the time scale of fixations. This is just one example of several.

3) Along these lines the figures are not optimal. While they show some images that look plausible, more and better labeled panels may be needed to explain the application examples, potentially in a protocol sketch for each figure, making it clearer which possible application (which attempted imaging resolution, scope, LM-EM correlation at subcellular or cellular scale, cell bodies or dendrites or axons etc.) is shown.

*Reviewer #2:*

Tsang et al. describe a hybrid chemical/freezing fixation method to results in specimens in an aqueous environment. The authors show this method allows subsequent enzymatic reactions, high-contrast en bloc staining, and correlative fluorescence microscopy. They also demonstrate the applicability in cultured cells, fly antennae, and acute mouse brain slices.

The methodological advance claimed by the authors is a rehydration step following freeze substitution. They compare their method to (van Donselaar et al., 2007), but there are uncited examples of hybrid protocols involving rehydration to achieve an aqueous environment for further tissue processing: Ripper, Schwarz and Stierhof, 2008; Stierhof and Kasmi, 2010.

Indeed, these studies demonstrate advances that Tsang et al. seem to be claiming as novel. For example, (Stierhof et al., 2010) explicitly show the ability to image GFP, YFP and RFP expression following rehydration.

I suppose the demonstration of high-contrast heavy metal staining and enzyme chemistry in an aqueous environment following freeze substitution is new, but not particularly surprising. Given the prior use of rehydration following freeze substitution for similar purposes, I doubt the presented method requires a new acronym (CCM).

Finally, one of the benefits claimed in the text and Figure 1 is the high temporal resolution of the method. For cell culture and fly antenna this is true but, in practice, this does not apply to the data presented from mouse brain in which chemical fixation by perfusion was first carried out.

*Reviewer #3:*

This paper highlights a new approach – CryoChem Method (CCM), which researchers can use to perform 3D electron microscopy on large specimens (up to a scale of ~500 µm in each dimension); these samples are first frozen rapidly at high pressure in their native state, and then freeze-substituted with mild fixation, before being rehydrated. This rehydration step is the key aspect of the work: it improves ultrastructural quality, while also importantly maintaining the viability of proteins that can be subsequently imaged by fluorescent tagging, e.g., DAB-based labeling schemes, such as miniSOGs and DAB polymerization before staining and embedding for SBEM, or other 3D imaging techniques. Although the CCM might seem like an obvious extension of existing approaches, it is not yet widely known by the scientific community, and is therefore likely to be of considerable value to structural and cell biologists.

Many valuable new techniques have some limitations in addition to their important advantages. Perhaps the authors could discuss some potential limitations (if they exist). For example, presumably some molecules that are retained in standard freeze-substitution protocols might be lost in the rehydration step. Is this a concern? One of the most important applications of the CCM approach is its ability to provide correlative LM/EM on large specimens. However, the LM is performed in the liquid state after hydration before embedding, whereas the 3D SBEM is performed after embedding. What is the precision of the correlative imaging? Does the rehydration introduce swelling or the embedding introduce shrinkage? The only figure that illustrates the correlative imaging is Figure 6, but it is difficult to assess the precision of the registration between the SBEM and the confocal LM images in this example. Could the authors provide some estimate of this?

---

## [Author Response]

[…] While the reviewers see the merits of this work, they raised a set of key concerns, and after discussion decided to request the following points for the revision of this manuscript:- Proper citation and discussion of the literature as indicated by reviewer 2; clarification of which additional advances were made compared to the published protocols.

We thank reviewer #2 for pointing us to the two important studies that demonstrated the preservation of fluorescent signals in cryofixed-rehydrated thin cryosections (Ripper et al., 2008 and Stierhof and Kasmi, 2010). We have now referenced both papers accordingly. We also clarified that we are not the first group to make fluorescence imaging possible in cryofixed-rehydrated tissues. Furthermore, in addition to van Donselaar et al., 2007, we now cited Dhonukshe et al., 2007 for the development of rehydration protocols for cryofixed samples.

As suggested by the reviewer, we also compared our approach to several existing methods. We described the key advances of our method in the Introduction and the Results sections. Although earlier methods permit fluorescent imaging in cryofixed-rehydrated samples, our method further enables (1) APEX2 labeling of target structures (2) and high-contrast EM staining in the same cryofixed specimen. (3) Importantly, our approach allows fluorescent imaging in large, whole-mount tissues, instead of 300-500 nm cryosections. Together, these critical advances allow one to perform 3D correlated light and electron microscopy (CLEM) in cryofixed tissues, as we demonstrated in the mouse brain (Figure 5).

- More extensive discussion of the limitations of the approach (see reviewers 1 and 3).

We agree with the reviewers that it is important to discuss the limitations of our method. In the revised Discussion, we now added caution remarks in a new paragraph.

…clarification for which examples the "improved temporal resolution" apply.

As reviewer #1 suggested, we removed “and high temporal resolution” from the Abstract to avoid potential confusion.

We also clarified in the Discussion that “*CCM can only improve the temporal resolution of biological events captured if the specimen is frozen in the live state, but not when the sample was first chemically fixed…*”.

…potentially problematic effects of the rehydration steps;

We addressed reviewer #3’s comment by including the sentence in the Discussion: “*Finally, there are also concerns that some molecules may be lost during rehydration if they are not properly fixed during freeze-substitution*”.

…tissue volume effects

We addressed this comment by noting “*To minimize volume artefact, epoxy resin is chosen because it causes minimal tissue shrinkage during embedding (<2%) compared to other embedding media (Kushida, 1962)*” in the Results. Since our CryoChem method (CCM) employs standard dehydration and embedding steps, we believe the degree of tissue shrinkage in CCM is comparable to that with standard EM protocols.

In addition, it is theoretically possible that the rehydration step could introduce swelling in CCM-processed tissues. However, we have not observed any obvious signs of swelling in the CCM-processed *Drosophila* antenna, when compared to the published antenna images that were similarly processed without rehydration (Shanbhag et al., 1999, 2000). In the Results section, we also noted “*…suggests that the rehydration step in CCM leads to little, if any, swelling in the antenna tissue*”.

Improve the descriptions of the reported applications. Which resolution of correlated imaging was achieved? How does this depend on the specimen and procedure?… A correlation at the level of subcellular structures while still being able to acquire large 3D EM volumes would be highly commendable, but it is not clear whether that is in fact achieved…

We thank the reviewers for these insightful comments. In response, we added 3 new images showing the correlation of a subcellular structure, heterochromatin (Figure 5—figure supplement 1), using our 3D CLEM protocol on a CCM-processed mouse brain. We note that this heterochromatin structure was not used as a fiducial marker. We also added a sentence to the Results section describing 3D CLEM: “*The high accuracy of correlation achieved by our 3D CLEM protocol is demonstrated by the successful alignments of multiple ultrastructures: the fine neuronal processes (Figure 5E) and a subcellular heterochromatin structure that was not used as a fiducial marker (Figure 5—figure supplement 1)*”.

In addition, we would like to point out that the primary objective of performing 3D CLEM on CCM-processed tissues is to enhance the quality of morphological preservation of the correlated structures. Although a better morphological preservation may help with image registration, the correlation accuracy is largely dependent on the correlation algorithm and the number of fiducial markers. To clarify this point, we added the comment “*so that the correlation can be achieved in optimally preserved tissues*” in the Results section.

How does it combine with circuit reconstruction?

One key advance of CCM is its compatibility with volume EM techniques, such as SBEM. As such, CCM is applicable to neural circuit reconstruction. We note that the SBEM volume of CCM-processed *Drosophila* antenna had a section thickness of 50 nm (Figure 2—video supplement 1 and Figure 3D). This z resolution is sufficient for circuit reconstruction.

Of note, we embedded CCM-processed samples in epoxy resin, which is amenable to common ultrathin sectioning techniques. If need be, one can reduce the section thickness below 50 nm.

In the Discussion, we also noted “*Furthermore, the ability to genetically label target neurons with fluorescent markers or EM tags in CCM-processed tissues can facilitate circuit reconstructions of identified neurons in optimally preserved specimens”.*

…include descriptions of the numbers on dataset size, resolution, etc. for the EM images and volumes presented.

As suggested, we added the pixel resolutions in the relevant Figure Legends for all TEM and SBEM images. For the SBEM volumes, we also added the SBEM imaging parameters (Z step size, Z dimension, raster size and pixel size) to the corresponding figure legends.

More and better labeled panels may be needed to explain the application examples.

We amended our manuscript according to the reviewers’ suggestions, as detailed below.

- We added two panels in Figure 2 to show additional images of CCM-processed specimen as compared with chemically fixed counterparts. Detailed descriptions of the morphological improvements are added to the corresponding Results sections and figure legends. We also included more labels for the EM structures shown in the images.

- We added a flowchart in Figure 3 to illustrate the steps for DAB labeling of target EM structures expressing APEX2. We also added more labels to the images.

- We added a supplemental figure to Figure 5 to demonstrate the correlation of a subcellular structure (heterochromatin) with our 3D CLEM protocol in a CCM-processed mouse brain.

- We added more details (e.g. excitation wavelength for DRAQ5 imaging, raster size and Z dimension for SBEM volume) to the Materials and methods section.